# Positron Annihilation Lifetime Spectroscopy as a Special Technique for the Solid-State Characterization of Pharmaceutical Excipients, Drug Delivery Systems, and Medical Devices—A Systematic Review

**DOI:** 10.3390/ph16020252

**Published:** 2023-02-07

**Authors:** Mariam Majida Shokoya, Beáta-Mária Benkő, Károly Süvegh, Romána Zelkó, István Sebe

**Affiliations:** 1University Pharmacy Department of Pharmacy Administration, Semmelweis University, 7-9 Hőgyes Street, 1092 Budapest, Hungary; 2Department of Nuclear Chemistry, Eötvös Loránd University, 1518 Budapest, Hungary

**Keywords:** positron annihilation lifetime spectroscopy, drug, dosage form, medicine, pharmaceutics, polymer

## Abstract

The aims of this systematic review are to explore the possibilities of using the positron annihilation lifetime spectroscopy (PALS) method in the pharmaceutical industry and to examine the application of PALS as a supportive, predictive method during the research process. In addition, the review aims to provide a comprehensive picture of additional medical and pharmaceutical uses, as the application of the PALS test method is limited and not widely known in this sector. We collected the scientific literature of the last 20 years (2002–2022) from several databases (PubMed, Embase, SciFinder-n, and Google Scholar) and evaluated the data gathered in relation to the combination of three directives, namely, the utilization of the PALS method, the testing of solid systems, and their application in the medical and pharmaceutical fields. The application of the PALS method is discussed based on three large groups: substances, drug delivery systems, and medical devices, starting with simpler systems and moving to more complex ones. The results are discussed based on the functionality of the PALS method, via microstructural analysis, the tracking of ageing and microstructural changes during stability testing, the examination of the effects of excipients and external factors, and defect characterization, with a strong emphasis on the benefits of this technique. The review highlights the wide range of possible applications of the PALS method as a non-invasive analytical tool for examining microstructures and monitoring changes; it can be effectively applied in many fields, alone or with complementary testing methods.

## 1. Introduction

PALS is a non-destructive radiological technique widely used in material science studies [1] since it is exceptionally sensitive to the free volume of the material [2]. The method is based on the lifetime and intensity of ortho-positronium (o-Ps) atoms in free volumes of given structures; positronium (Ps) trapping in vacancies has a characteristic lifetime and, therefore, allows one to measure the size of a vacancy cluster, providing information about the structural characteristics of solids, liquids, and semi-solid materials [1,3,4]. PALS is utilized in several sectors for the examination of energy carriers [5,6,7,8], catalysts [9,10,11], and packaging materials [12,13] and for interrogating defects and pores in metals [14,15], ceramics [16,17], and polymers [3]. The principles of positron annihilation in condensed matter have been extensively discussed [18,19]. A brief outline of the physical basis of the process is that this technique employs the positron as a probe. The positive electric charge is repelled by the ion cores and becomes localized preferentially in any atom-sized part of the material, making it ideal for discerning the characteristics of voids and defects in this size range [4,19].

The source of the positrons, typically placed between two identical samples of a material under investigation, is usually a radioactive isotope, such as Na-22 [19]. Na-22 decays into Ne-22 through the β^+^ decay process, and an energetic 1.28 MeV gamma ray is emitted within a few picoseconds after positron creation, which serves as the birth signal for the positron [4,19]. Part of the anti-matter portion of the physical universe, the positron, is the antiparticle of the electron; the positron emitted by the radioactive source and the electron from the media collide [4,19]. Thus, when an energetic positron is injected into a sample, it undergoes a series of inelastic collisions with its surroundings to arrive at thermal equilibrium, after which the thermalized positrons may undergo free annihilation with electrons within the material at a rate that depends on the electron density of the medium, with typical lifetimes of 100–500 ps [3], or form a quasistationary state called Ps [19]. The electron–positron annihilation process occurs in different modes depending on the mutual orientation of the spins of the particles participating in the annihilation process, and each mode has a characteristic lifetime (Table 1.) [4]. Ps is found either in the spin singlet state and called para-positronium (p-Ps), or in the triplet state and called o-Ps [19]. 

In the matter, o-Ps accumulates in regions of low electron density and thus occupies the core of the voids of the material (the wall of the core is formed by the electrons from neighbouring molecules). It is annihilated via collisions with electrons from the surrounding media by so-called “pick-off” annihilation [3], and two 0.511 MeV annihilation photons are created [1], conforming to the matter–energy equivalence principle. The parallel spinning positron is “picked off” by annihilation with an anti-parallel spinning electron from the surrounding wall, and thus the positron is annihilated not by the electron to which it is bound, but by an electron in its neighboring environment [4]. Therefore, by measuring the annihilation lifetime (i.e., the survival time of the positron in the medium), it is possible to directly obtain the electron density that is encountered by the positron in the medium [19]. The o-Ps lifetime is a direct measure of the proximity of the Ps to the local environment, as the bound electron of the Ps is repelled by the electrons in the medium [3]. The elapsed time between the initial production of the positron (1.28 MeV gamma ray) and the detection of one of the two annihilation photons (0.511 MeV gamma ray) is therefore a measurement of the positronium lifetime in the material under investigation [1,4,19]. Thus, for the detection of the lifetime decay curve, two gamma ray detectors (BaF_2_ or plastic scintillators) are tuned to detect the two energy levels (1.28 MeV for the start and 0.511 MeV for the stop) and are connected to a time counter capable of resolving events occurring nanoseconds apart [4,19]. The o-Ps lifetimes and intensities are functions of the electron density at the annihilation site. Lifetime measurements can discriminate among the different lattice locations at which the positron annihilates and consequently give information about open-volume defects [18]. O-Ps lifetimes and intensities have been associated with the size of free volume sites and the probability of formation or the number of free volume elements present, respectively [3]. The free volume is divided between a number of sites in the average free volume accidentally occupied by Ps atoms, which are then quenched due to pick-off annihilation [19], and the lifetime of an o-Ps atom depends on the size of the free volume in which it is located [2]. The simple model of a Ps atom in a spherical potential well of radius (R) shows a correlation between the o-Ps lifetime and R [2,19,20]. The correlation between the o-Ps lifetime (τ) and R relates to the spherical free volume of cavities in molecular materials and is derived from the Tao–Eldrup equation [21,22,23,24]:(1)$$\tau =0.5{\left[1-\left(\frac{R}{R+\Delta R}\right)+\frac{1}{2\pi }\sin\left(\frac{2\pi R}{R+\Delta R}\right)\right]}^{-1}$$
where ΔR is the fitting parameter, used as a measure of the electron layer thickness around the free volume cavity—empirically, ΔR = 1.656Å. 

The PALS method typically relies on an analog coincidence measurement setup and allows one to estimate the positron lifetime in the material sample under investigation [1]. The spectrum of the positron lifetime is obtained as a histogram of counts [N(t)] (number of measured events) versus time; the resultant time spectrum is a sum of exponentials of lifetimes in a medium [19] (Figure 1). In recent years, digital oscilloscopes and fast digitizers have replaced traditional analog timing modules in PALS experiments, and digital signal processing has become another important factor affecting the time resolution besides the scintillator properties [1]. The resolution function, which is the response of the spectrometer to prompt coincidence events, has a Gaussian-like shape and is characterized by its full width at half maximum; τ is the inverse slope of the o-Ps component, and the intensity (I) is the area under the o-Ps slope. PALS data analysis can be achieved by either direct deconvolution of the spectrum with programs employing various regularization strategies (e.g., CONTIN, MELT, RESOLUTION) or by fitting a theoretical model to the experimental data with spectra-fitting programs (e.g., PositronFit, PALSfit) [3,19].

Among the numerous analytical techniques used in material irradiation studies, PALS is well known for its spectacular sensitivity to atomic-scale vacancy-type defects, and it is of great importance for microstructural investigations of some technologically important materials. The correlation of PALS data with independent theoretical predictions of free volume and the usefulness of free volume data in quantifying the physical and mechanical properties of various materials with different consistencies [4] make this technique popular in the characterization of metals, alloys, semiconductors, ionic crystals, insulators, polymers, and other substances strongly related to the physics and chemistry of solids [18,25,26,27]. A well-known method related to PALS, based on the same physical phenomenon, is positron emission tomography (PET), a radiation-based medical diagnostic tool of functional imaging extensively used in the fields of oncology, neurology, and cardiology [28,29,30]. Similar to PET and complementary to its applications in material sciences, the other outstanding field of application of positron annihilation is life sciences, including broad research categories such as drug delivery systems, biocompatible materials, macromolecular and membrane studies, and biological tissue research [30]. 

Amorphous systems are widely used in the pharmaceutical industry to increase the solubility of Biopharmaceutics Classification System (BCS) II and IV drug substances, increase bioavailability during drug metabolism, and stabilize incorporated drug substances. The PALS method is effective primarily in the investigation of partially or completely amorphous systems for the supramolecular characterization of drug delivery systems and medical devices. It is capable of analyzing the microstructural properties of studied systems, such as atomic-size free volumes, defects, and monitoring changes related to external factors. The effects of storage conditions, humidity, and temperature all influence the stability and release of the active ingredient and determine the mechanical properties of the matrix. Physical ageing in polymeric systems and enthalpy relaxation are often probed by accelerated stability tests. PALS is applied as a microstructural tool to study contact lenses and intraocular implant materials in relation to oxygen permeability and the diffusion of certain ions and water in the matrix. 

However, pharmaceutical technology, as a frontier field of materials science, has not yet introduced the PALS technique into the realm of industrial drug research and development. This can be attributed to a number of factors, such as certain drawbacks in the PALS method’s performance, confounded by differences in the confidence of results, the complex experimental configuration, and varying radiation source configurations and data analysis procedures [3]. The need to achieve even better precision or supplemental information is constantly triggering new developments in the method (improvement of time resolution, count rate, simplification, and stability of the system) [31]. However, the elimination of the disadvantages has not yet reached a level of development that would allow PALS method to gain ground in such a highly regulated field as pharmaceuticals. Furthermore, it can be used as a complementary analytical method [3] in fundamental research, and its importance is growing with the increasing use of nanotechnology in the pharmaceutical industry. Highlighting the role of PALS in polymer characterization and the application of these materials in conventional pharmaceutical formulations, novel drug delivery systems (e.g., suspending, coating, and drug release controlling agents), packaging materials, and medical devices, there is evidence that with the proper selection of the surface and bulk properties of polymers through the application of a wide range of analytical methods, including PALS, pharmaceutical development [32,33] can be achieved. Therefore, to provide a comprehensive picture of the current state of PALS applicability in the frontier field of pharmaceutical technology, a systematic literature review has been carried out. To the best of our knowledge, the PALS method represents a special approach to the solid-state characterization of pharmaceutical excipients, drug delivery systems, and medical devices; thus, an in-depth examination of its application could extend the basic drug development perspective. The aim of this study was to summarize the technique’s potential utility in pharmaceutical investigations in order to identify and describe the drug delivery systems and medical devices in which the PALS method can be informatively applied. Furthermore, as PALS could play a complementary role in addition to the commonly implemented analysis tools (e.g., NMR, DSC, SEM, XRD, small-angle X-ray scattering) [3,18,34], the testing methods used in research in combination with PALS are reported, and the data types extracted from the parallel examinations are presented.

## 2. Results

### 2.1. Database Search and Included Studies

The search terms yielded a total of 3936 matches from the four databases—more precisely: 373 from PubMed, 594 from Embase, 2619 from SciFinder-n, and 350 from Google Scholar. The flowchart (Figure 2) below summarizes the individual steps of the publication selection and screening process. After the elimination of duplicates and the consideration of the eligibility criteria, a total of 129 articles were selected, and after the complete study of the scientific materials, 94 articles were ultimately used in the data processing.

### 2.2. Results of Studies

The objectives of the scientific publications (more specifically the substances (active substances, excipients, and processed excipients), drug delivery systems (polymeric films, micro- and nanofibers, lyophilized formulations, tablets, transdermal patches, intrauterine delivery systems, and nanostructured delivery systems) and medical devices (contact lenses, intraocular lenses, and dental fillings), the aims of the studies, and the functionality of the PALS method compared to other structural analysis methods are all briefly summarized in Table 2. 

#### 2.2.1. Substances

##### Active Pharmaceutical Ingredients

PALS was applied to examine the microstructure of active substances and supramolecular structures. Verapamil hydrochloride’s dependence on temperature was characterized through glass transition temperature (T_g_) and free volume measurements. The hole density and fraction derived from PALS were correlated via pressure–volume–temperature (PVT) experiments [35]. In the examination and microstructural characterization of the Metformin drug and its antimicrobial complexes with V(VI) and Cr(III) ions, the PALS spectra revealed two lifetime components in the complexes attributed to their homogenous, crystalline structure. The relatively small differences between the spectral parameters of the complexes were caused by differences in the molecular weight and molar ratio of the metal ions [36]. The investigation of the structural characteristics of a series of N-heterocyclic compounds as organic solids was performed via PALS measurements. The o-Ps lifetime analysis provided data about the structure and electron donation characteristics of the N-atom containing compounds. PALS spectra showed large variability in these parameters due to the absence or presence of certain functional groups. [37]

The free volume of Olanzapine, as well as the effects of temperature and pressure related to the manufacturing process of its active substance, were investigated. The polymeric form I was irreversibly changed by temperature while the o-Ps lifetimes increased, and after further heating, they decreased. Polymeric form II showed non-monotonous changes caused by heating, and the crystalline structure recovered. As for pressure changes, the dense structure of the atomic rings precluded structural collisions [38].

##### Excipients

The aging effects of two types of PEO polymers used as tablet binders, coaters, and drug-release modifiers were tracked by the PALS method during accelerated stability testing. Due to the structural relaxation under the stress conditions of the two PEO samples, the free volumes of the holes decreased as the polymeric chains became more ordered. Furthermore, moisture exerted a plasticization effect and facilitated the recrystallization of the samples [39].

##### Processed Excipients

The relationship between the physical properties and supramolecular structure was examined in microcrystalline cellulose (MCC)/isomalt composite pellet cores. Pure MCC had higher o-Ps lifetime values, while the presence of isomalt polyol decreased the free volumes and the presence of ordered MCC chains [40]. PVP was used as the binder, the nonionic stabilizer was pelletized, and the PALS spectral characteristics were determined by different descriptive models [41]. The mebendazole-containing extradite was prepared and contained soluble polymeric excipients with sulfobuthylether-β-cyclodextrin, PEG 6000, and plasticizers. Furthermore, a four-week stability testing was carried out to examine the drug release stability of the samples. Hot melt extrusion ordered the polymeric chains and decreased the o-Ps lifetimes compared to the physical mixtures. In both systems, the PALS method revealed low o-Ps lifetime values, which are related to the crystalline state [42].

The microstructure and the effects of water and sucrose contents in spray-dried polymeric blends of hydrophobically modified starch (HMS) were analysed. Phase separation was observed, and two phases were generated. Similarly, in the previously examined systems, water acted both as an antiplasticizer and a plasticizer depending on its concentration. When the sucrose concentration increased, sucrose exerted an antiplasticization effect and decreased the free volume [43].

#### 2.2.2. Delivery Bases and Drug Delivery Systems

##### Free and Drug-Loaded Polymer Films

The depth profiles of the free volumes of drug delivery systems, specifically polymeric films of the mixture of PVP/PEG and copolymeric systems of PVP-PEGDA/PVP-PEGMMA, were investigated by PALS. Significant variations in the PALS spectral parameters were observed; the polymeric mixture with hydrogen bonding between PEG chains had larger free volumes and wider distributions compared to the copolymeric networks. Longer o-Ps lifetimes were measured near the surface compared to the bulk of the polymeric material [44]. Microstructural changes were tracked by real-time PALS during the phase transition of the gel drying to a polymeric hydrocolloid film. The addition of Carbopol 71 G to the system in different concentrations affected the measured characteristics. At lower concentrations, Carbopol 71 G was incorporated into the SA matrix; after reaching saturation (0.2 w/wt. %), Carbopol 71 G created a new phase and slowed the drying rate [45]. Ethylcellulose and PVP blends, used in solid dosage forms as coatings and drug-release controllers, were studied. Microstructural analysis revealed that increases in both the free volume hole size and fraction were caused by increasing the ethylcellulose concentration. The combined structural properties of the polymeric blend determine drug permeability and, therefore, drug release properties [46].

Sucrose palmitate is a potential enhancer that can be used orally. Buccal polymeric films were prepared with different sucrose palmitate concentrations, two types of polymers (HPMC-5 and HPMC-15), and different preparation temperatures. The recommended concentration for sucrose palmitate is 1%. Similar results were generated in the microstructure, independent of the sucrose palmitate concentration and polymer concentration [47]. The structural analysis and comparison of Eudragit dispersions and films, and the effects of the dibutyl sebacate (DBS) plasticizer were examined by the PALS method. By increasing the DBS concentration in the Eudragit free films to 10%, the hole size also increased, and above this concentration, there were no significant changes. The instability of the Eudragit coatings was related to the increased molecular mobility caused by the plasticizer [48]. The effects of low-molecular-weight polyols (glycerol and sorbitol) on the physical aging of gelatine films were studied. Polyols enhanced molecular packing, which is governed by the complex interaction between gelatine, water, and the plasticizer, rather than a reduction in the free volume measured by PALS [49].

The effects of the water content on the structure of the amorphous carbohydrate matrices of maltopolymer–maltose blends were analyzed. In the glassy state, at low water contents, water acted as an antiplasticizer and decreased the free volumes of holes by filling them, but at higher concentrations, it acted as a plasticizer and increased the hole volumes by interacting with the hydrogen bonding network of the carbohydrate matrices [50]. The effects of water on the structure and physical properties of amorphous maltodextrin–dextrose blends were examined by PALS and PVT experiments. The system was strongly dependent on the temperature and water activity in both the glassy and rubbery states. At a low moisture content, the density and hole size increased with water activity due to the swelling of the matrix. If the water activity was further increased, both the density and degree of swelling would decrease [51]. The microstructures under different conditions of the maltopolymer and maltose blends were examined by the PALS method in PVT experiments. Both the microstructure and specific volume were analyzed in glassy and rubbery phases. The free volumes of holes were linearly dependent on the temperature. When the maltose concentration in the blend was increased, the free volume decreased due to denser molecular packing. Water exhibited a complex concentration-dependent, as in previous research—it can act as both a hole-filler and a plasticizer [52].

The plasticization effect of water was investigated in polymers used for tableting, and a 30-day stability test was performed to track the physical aging of the system. Samples stored at 65% RH and 75% RH underwent a two-step swelling process; initially, water formed hydrogen bonds and crosslinked polymeric chains. After 30 days, excess water generated larger cavities [52]. The effect of the PEG 400 plasticizer on the aging of methylcellulose, i.e., the metolose polymer used as a film-forming material in tablet coating, was examined. The one-phase polymer matrix swelled and the free volume decreased, but with increased concentrations of PEG (above 20% *w*/*w*), a two-phase structure was formed, and the additional PEG helped with the uptake of water while the free volume did not change [53]. The interactions of PEG 400 with metolose films were studied at increasing concentrations during accelerated stability testing. At lower concentrations, the PEG was incorporated into the polymeric chains, but above 50 w/wt. % PEG content, droplets were formed within the matrix. PEG acted as a plasticizer, increased the free volume size, and decreased the interaction between metolose films [54]. The free volume in the gelatine–glucose system was investigated under different conditions, and the void volume’s dependence on water loading, drying, and pressure was investigated and tracked by PALS. The addition of glucose to the system led to the formation of crosslinks. Compared with the superdry gelatin–glucose sample, saturation with H_2_O resulted in the creation of nanovoids and the swelling of the system [55].

The effects of polymer chain length and selected composition on free volume holes were investigated in PVP polymeric films. The effect of the PVP chain length was investigated on the recrystallization of amorphous bicalutamide in polymeric thin films. PVP K90, with the longest polymeric chain length, exerted the weakest inhibitory effect on recrystallization, while PVP K30 exerted the greatest inhibitory effect. The optimal chain length should be selected for drug stabilization, and further physicochemical factors should be considered [56]. Amorphous spray-dried polymeric dispersions were prepared with indomethacin and ketoconazole drugs, and the effect of polymer selection was examined (PVP, PVP-VA, HPC, and HPMC-AS) on the free volume. The flexibility (polyvinyl-based polymers) versus inflexibility (cellulose derivatives) of polymeric chains, i.e., backbone chain rigidity, greatly impacted the free volume, while cellulose derivatives formed larger free volumes due to chain restrictions [57]. The developmental and structural study of PU-based nail lacquers showed a correlation between the microstructure, density, and the sustained release of terbinafine hydrochloride from dried polymeric films. The polymeric composition of nail lacquer crucially influenced the diffusion of the drug. Nail lacquers had unusually high free volumes of holes in relation to the sustained release of the drug [60]. The microstructural and binding properties were examined in molecularly imprinted PE/PP non-woven fabrics with penicillin G. Penicillin G’s molecular diameter (0.500 nm) perfectly fits into the free cavity size diameter (0.507 nm), i.e., the binding sites, of the non-woven fabrics [59]. The plasticizer effects of polyols (glycerol and xylitol) were examined in a lidocaine base containing mucoadhesive polymeric films made from HPC. The two differently molecular-weighted polymers did not differ significantly in terms of free volume, but the polyols had different effects on the matrix structure. Xylitol decreased the free volume size as it filled the voids, while glycerol increased it because it destroyed the H-bonds in the network [58].

Solvent-cast and freeze-dried polymeric films containing vitamin B_12_ were prepared, and accelerated stability tests were performed. A correlation was found between the parameters of the drug release profiles and the o-Ps lifetime values. Increases in the Carbopol 71G content in SA/Carbopol 71 G composites, the freeze-drying process, and storage at 75% RH all decreased the drug release from the matrix. After four weeks, larger free volume holes indicated the presence of extended polymer chains under elevated relative humidity conditions. The extent of swelling at 75% RH was lower in freeze-dried films than in cast films [61]. The structures of the Eudragit polymeric films (Eudragit^®^ L 30D-55) used in the coating liquids containing diclofenac sodium were investigated with variable drug contents, and the effect of water was assessed during storage. Water uptake during stability testing affected the microstructure, and water molecules formed new links with the polymeric matrix with plasticizing effects [62]. The moisture sorption of ketoconazole- or indomethacin-loaded spray-dried dispersions with different matrices, containing four types of polymers (PVP, PVP-VA, HPC, and HPMC), was measured by PALS, and the effect of humidity on the free volume hole change was tracked. Polymers, active substances, and humidity levels all significantly affected the free volume of the solid dispersions, while drug–polymer interactions inhibited recrystallization, but exposing the system to humidity levels higher than 75% led to instability issues [63]. The structure and temperature responsiveness of drug-incorporated (sulfamethoxazole or paclitaxel) PU membranes were analyzed. The synthesized system contained a fixed and a reversible phase, which could be “switched” on and off by temperature changes. The PALS method highlighted dramatic increases in the free volume size and fraction upon heating from physiological temperature to 44 °C due to the sharp phase transition from the glassy to the rubbery state. The drug release was proportional to the duration of heating, and drug diffusion could be switched off by cooling the system [64].

##### Neat and Loaded Micro- and Nanofibers

Placebo and drug-loaded supramolecular holes are trackable using the PALS method. The PALS method revealed a significant difference between the placebo and nebivolol hydrochloride-loaded fibers. Drug-loaded PVA had smaller free volumes, which adheres to the “hole-filling theory”. Drug release was complete and independent of pH [65]. Iodine-containing topical nanofibers were successfully prepared. The PALS results revealed changes as a function of different polymeric ratios of PVP/PVP-VA and the presence of iodine. The iodine drug showed reduced hole sizes, while hole filling resulted in decreased mechanical strength. The presence of PVP-VA ordered the polymeric chains, resulting in a narrow o-Ps lifetime distribution and thus larger and more uniform hole sizes [66]. Free volumes of papaverine hydrochloride-containing nanofibers prepared from HPC/PVA composites were tracked during the preparation process. The degree of fiber formation and possible H-bonding between the drug and polymers organized the polymeric chains and decreased the o-Ps lifetimes. The applied polysorbate 20 solubilizer plasticized the matrix in both, the physical mixtures and the fibers [67]. The applied technology and parameters influenced the structure of the fibers. The PALS method was applied in the investigation of the structural properties of the intermediate products of directly compressed tablets containing vitamin B_12_, prepared from the milled nanofibrous matrix and cast-free films. The micronization process did not significantly impact the free volume distribution of the nanofibers but did reduce the free volume of the films [68]. 

Polymer composition, concentration, and the solvent effects of alcohol and water were examined in PVP/PVP-VA-based polymeric nanofibers. Water acted as a plasticizer; it reduced free volumes compared to alcohol, which did not act as a plasticizer [69]. PVP-based homopolymer and copolymer systems made with PVP-VA were produced by the developed high-speed rotary jet device, and samples were morphologically analyzed to infer the robustness of the equipment. The PALS method revealed a correlation between the mechanical properties and free volumes in the examined RPM range, while the compact design and controllability of the operational parameters enabled the formation of aligned orientation nanofibers [70]. Buccal fenofibrate-loaded electrospun fibers and physical mixtures prepared from the PVP carrier matrix were examined by PALS. Polysorbate 80 solubilizer was selected to increase the solubility of the drug in an alcoholic PVP solution. Polysorbate 80 increased the free volume in both sample types and exerted a plasticization effect on the polymeric matrix. The use of the excipient could be favorable for reducing matrix rigidity [71]. The microstructure of the fibrous elements in the nanofibrous PVA sheets and PVA powder containing colistin sulfate was tracked by the PALS method. The heat treatment of PVA fibers resulted in a more ordered structure with decreased free volumes, which could thus function as a permeable rate-controlling membrane for the drug. The alternative layer design is capable of controlling drug release in the topical application area [72].

The crystalline to amorphous transition of carvedilol during the electrospinning process was tracked by the PALS method. The fibers had a large stress tolerance capacity; carvedilol was only partially crystallized over the four weeks of accelerated stability testing. Progressive changes in the supramolecular structure were found during storage, while the free volumes of holes were increased by the altered orientation of polymeric chains [73]. The phase transition of the active substance papaverine hydrochloride occurred, and major morphological alterations were observed in HPC/PVA composite systems. A two-step aging process of the polymeric matrix was revealed by the PALS method. After one week, the elevated temperature affected the molecular mobility of the polymer chains, resulting in the relaxation of the matrix. After two weeks, a new structure was formed, possibly involving hydrogen bonds between polymer chains. Longer exposure (two to four weeks) led to different structural organizations due to the destruction of hydrogen bonds [74].

The effects of different solubility enhancers (triethanolamine and pH modifier sodium hydroxide) were examined in PVP and HPC nanofibrous matrices and physical mixtures during accelerated stability testing. In the furosemide-containing fibrous systems, triethalonamine acted as a plasticizer and increased the o-Ps lifetimes compared to the NaOH-containing matrix, which was more densely packed [75]. Excipients were added to amorphous solid dispersions to improve aqueous solubility and physicochemical stability, and the effects of the polysorbate 80 surfactant and hydroxypropyl-β-cyclodextrin were investigated on the molecular packing of the drug-free and metoclopramide-loaded buccal nanofibers. Both excipients allowed the generation of a uniform nanofibrous matrix; polysorbate 80 acted as an external plasticizer, while cyclodextrin was an internal plasticizer and molecular packing enhancer [76].

Next, 3D-printed PLA-based polyvalent test plates (PVTP) that were surface modified with oligoamines were prepared, and the PALS method was used for microstructural analysis. Approximately 100 µm of the system’s surface was analyzed by the PALS method, and chemical changes were tracked in the supramolecular range. Surface-modified PLA PVTPs had significantly shorter o-Ps lifetimes and fewer free volume holes compared to the unmodified samples because amines formed in between the polymeric chains and polymer molecules in the filaments [77].

A bacteria-embedded nanofibrous web was successfully prepared. After one week of storage under stress conditions, the embedded microorganisms increased the free volumes of holes; this is associated with the metabolic product of carbon dioxide produced by *S. maltophilia*, which could enhance the polymeric matrix. After three weeks, we saw an expansion caused by the collision of the fiber web [78].

##### Freeze-Dried Formulations and Biologics

The PALS method is suitable for the study of the structural properties of freeze-dried sodium hyaluronate. Due to the presence of phosphate salts in the formulation, the freeze-dried samples were densely packed and ordered; the salt interacted with the polymeric matrix, which resulted in shortened o-Ps lifetimes [79].

The effects of glycerol and water excipients on the microstructure of freeze-dried maltooligomer matrices were analyzed by PALS. Glycerol acted as an antiplasticizer in the studied range (0-20 wt. %) and reduced the average molecular hole size. Depending on the level of hydration, water acts as a plasticizer and an antiplasticizer [80]. Freeze-dried powders of maltodextrin carbohydrate with different concentrations of glycerol and water were prepared to study the effects of these excipients in the glassy state. Glycerol reduced the average molecular size, which improved the performance of biostabilization. In the glassy state, water at different concentrations exerted antiplasticizer and plasticizer effects on the system [81]. The effect of sucrose was analyzed in freeze-dried sucrose and starch blends. The addition of sucrose plasticizer had a strong effect on the free volume and Tg [82]. 

The effects of sucrose and trehalose stabilizers and the further addition of polyols (glycerol and sorbitol) to hydroxyethyl starch (HES)-based lyophilized formulations containing human growth hormone (hGH) were examined. The system was analyzed in terms of density, molecular mobility, and free volume, and the results were evaluated and correlated qualitatively. Sucrose appeared to be a better stabilizer compared to trehalose because it reduced the free volumes of holes more effectively [83]. The effects of alkali chloride electrolytes on freeze-dried sucrose-based protein (bovine serum albumin and recombinant human serum albumin) formulations were studied. PALS measurements can offer good predictions of storage stability, but the decrease in the free volume did not directly correlate with the impact of the salts on stability. It was concluded that small amounts of LiCl and NaCl significantly stabilized the systems [84].

##### Tablets

The PALS method is a useful tool for tracking structural transitions during storage. The free volumes of PVP binder-containing tablets with theophylline were characterized during a 30-day stability test under different humidity conditions. Based on the results, a good correlation was found between the relative humidity during storage, drug release, and the free volumes of holes in the system. The PVP pore structure was rearranged, and the water associated with the increased humidity (65% RH) induced the glassy to rubbery state transition of the polymer, which significantly affected theophylline release [85]. The effects of the absorbed water on the microstructure and drug release of the famotidine-containing polymeric matrix (PVP or Carbopol 71G NF) tablets were examined during accelerated stability testing. In the case of the Carbopol-based matrix tablets, during the four weeks of testing, the volumes decreased due to aging. In contrast, the PVP-based tablets showed continuously changing o-Ps lifetimes during storage, which were related to the concentration-dependent dual effects of water on the system [86].

##### Transdermal Patches

The correlation between metoprolol tartrate release from transdermal patches and the free volume of the polymeric matrix (hypromellose and methylcellulose blend) was examined. Drug-loaded patches exhibited an intermolecular interaction that was correlated with the drug release dynamics. PALS served as a method of tracking drug-free and drug-loaded systems and supported drug release examinations [87]. Zolmitriptan-containing polymeric adhesives were examined. The sizes of free volume holes were greater in amide adhesives than in pyrrolidone adhesives. The release of zolmitriptan from the acrylate adhesive systems was controlled by the drug–adhesive interaction rather than the free volume properties of the adhesives [88]. The influence of the polymeric blended microstructure of two types of HPC (Metolose SM 4000 and 90 SH 100.000 SR) and Eudragit NE 30D on the metoprolol tartrate release from the transdermal patch was investigated with the support of PALS. The functional groups of the polymer formed H-bonds during water sorption and increased its volume [89].

##### Intrauterine Delivery Systems

The PALS method sensitively monitored the changes in the levonorgestrel hormone-releasing intrauterine system over five years. After three years of in vivo application, incrustations formed on the surface of the system, and these crystalline deposits affected positronium formation. The growth of the inner core was attributed to the physical aging of the core material, which may have serious consequences for the applicability of the system [90].

##### Nanostructured Delivery Bases and Drug Delivery Systems

The inner voids of rigid polyphenylene dendrimers were examined. Powder samples were pelletized for the PALS experiment. The results show that the sizes of the inner voids of the dendrimer branches, which can incorporate guest molecules, increased with the sizes of the molecules (as the number of dendrimer generations (Gx) increased). After the third generation (G3), these voids could not increase further due to the “overlap” of the large dendrimers [91]. PALS was used as a supportive method to analyze the structures and defects of nanoceria, a hybrid nanostructure synthesized with cerium dioxide (CeO_2_) and microcrystalline cellulose. The results reveal the low intensity of o-Ps formation because almost all of the positrons were annihilated in the CeO_2_ quantum dots; this mechanism is essential in scavenging reactive oxygen species (ROS). Furthermore, the effects of the pH and temperature parameters on the manufacturing process were tracked [92]. ZnO/chitosan biocidal nanocomposites were prepared against *Pseudomonas putida*, and hole generation related to positron irradiation by PALS was investigated. In this case, defects and the interfacial charge migration were analyzed by tracking the *τ*_2_ component, as this indicates the interaction between ZnO and chitosan. Chitosan acted as an electron acceptor and trapped the electrons ejected from ZnO generated by the high-energy positrons [93].

Radiotracers were applied in order to characterize and pre-screen the binding properties of porous materials, more specifically, hollow silica shells. The absorption of silica shells was affected by the size, charge, and hydrophobicity of the micropores; the size and porosity of the shells were measured by the PALS method [94].

#### 2.2.3. Medical Devices

##### Contact Lenses and Intraocular Lenses

PALS is an effective method used to probe the microstructures of biotechnologically relevant materials. The free volumes of holes in hydrogel (Omafilcon A) and silicone-hydrogel (Comfilcon A) lenses were studied. The results showed that the free volume size and fraction correlated with oxygen permeability. Silicone-hydrogel lenses had an oxygen permeability that was several-fold greater and only a slightly lower water content, which together provided greater wearing comfort to patients [95]. The PALS method clearly differentiated between the free volume sizes and fractions of soft (Etifalcon A and Narafilcon A) and rigid gas-permeable (Fluor-Silicon-Methacrylate-Copolymer) contact lenses. Contact lenses with higher oxygen permeability coefficients and a lower water content had more and larger free volume holes [96]. Void size and free volume holes were examined by PALS in three types of hydrogel and silicon-hydrogel contact lenses, namely, monthly Balafilcon A lenses and daily Hilafilcon A and Polymacon lenses. The void sizes arising in oxygen diffusion through the system are crucial, and oxygen diffusion is strongly related to the volumes of the holes [66]. The free volumes of different hydrogel (Ocufilcon D, Etafilcon A, and Omafilcon A) and silicone-hydrogel (Narafilcon A, Enfilcon A, and Comfilcon A) contact lenses were examined. Depending on the material quality, significant differences arose in the parameters of the PALS spectrum; silicone-hydrogel lenses exhibited greater volumes and thus greater oxygen permeability [98].

Polymeric hydrogel contact lenses (five samples from the Proclear family, CooperVision) made using phosphoryl choline technology were investigated regarding their microstructure and defects. The PALS method was used to characterize free volume vacancies in the hydrogel lenses, as well as the annihilation of Ps in point defects. The mean defect sizes were reflected by the *τ*_1_-*τ*_2_ parameters, which differed according to modifications in the preparation technology [99]. The free volumes and defects of hydrogel (Proclear family) and silicone hydrogel (Bioinfinity family) contact lenses prepared using phosphoryl choline (PC) technology were investigated by PALS. Based on the PALS spectra, hydrogel lenses had greater sizes and quantities of free volumes compared to hydrogel lenses, which correlated with increased oxygen permeability and comfort [100]. The structural investigation and defect analysis of soft contact lenses performed by PALS highlighted the lower degree of disorder in the hydrogel contact lenses due to their water-filled macropores compared to the silicone-hydrogel lenses. The geometry and nature of the defects completely differed due to the different preparation technologies applied for the lenses; hydrogel lenses exhibited more compact structures [101].

The effects of glucose and water in Fluoroperm 92 were investigated; this is an important consideration for diabetic patients wearing lenses. The plasticizer effect of water manifested in increased volumes after the swelling process, while the presence of glucose further helped enhance sorption and decreased the volumes due to the filling of the holes [102]. The effect of X-ray irradiation on soft contact lenses (Narafilcon A) was examined. An increased total irradiation dose decreased the free volume size and slightly increased the number of free volumes. The X-rays did not affect the microstructure in a negative way and did not cleave monomer bonds [115]. Calcium deposits were investigated in PHEMA soft contact lenses (Soflens and Super Soft). Calcified lenses showed decreased o-Ps lifetimes, while Ca^2+^ ions diffused and localized in the cavities of the system. Lens spoliation altered the optical transparency and microstructure of the lenses [103].

The microstructures of vinyl-based polymers used as intraocular lenses (IOL) were investigated via the PALS method. The IOLs were synthesized from MMA, BMA, and EHA monomers, with or without EDGMA crosspolymers. The lifetime spectra revealed that BMA-based lenses surpassed MMA-based ones in terms of free volume size and fraction due to the larger butyl functional group and longer chain of BMA, which increased the distance between the polymeric chains. The use of higher concentrations of EDGMA crosslinker decreased the free volumes and drug diffusion coefficients [104]. Commercially available intraocular implants, Alcon lens models, and SA60AT and SN60AT hydrophobic lenses were characterized by PALS. The base materials were crosslinked with PEA and PEMA. The structure of SN60T modified with a blue light chromophore was molecularly dense, manifesting decreased o-Ps lifetimes, free volume sizes, and fractions [105].

Silicone oil with a base of polydimethylsiloxane is sometimes applied in ophthalmology along with intraocular lenses. In the study, hydrophilic PHEMA and the impact of hydrophobic modification in the two types of hydrogel-based intraocular lenses were examined. After 6 months of incubation in an oil medium, PALS revealed that the silicone oil did not penetrate the internal structure of the implant and only interacted with the surface of the lens [106]. Polydimethylsiloxane (PDMS)-based silicone oil impacted the structure of the intraocular lenses; for this, PMMA rigid lenses and PHEMA flexible lenses were investigated. Hydrophilic PHEMA-based lenses absorbed the silicone oil; the swollen state of the matrix was reflected in the smaller free volumes, while hydrophobic PMMA lenses did not absorb the excipient and acted as a barrier to the oil [107].

##### Dental Fillers

Acrylate materials affected by different durations of polymerization (light exposure) were examined via long-term stability testing. The degradation of primary free volume holes after 2 years increased the number of holes due to the aging process, and this may also affect the mechanical properties of the material [108]. Light-cured dimethacrylate-type dental restorative composites (DRC) were studied in the initial and strongly light-cured states via the PALS method. The polymerization volumetric shrinkage under light-curing was more efficient in the Charisma^®^ nanocomposite than in the Dipol^®^ nanocomposite [109]. The microstructures of four dimethacrylate-based dental composites and the effects of curing light intensity were investigated by PALS. The lifetime components of the spectra differed for all four materials, and the sizes of the free volume holes depended on the irradiation time. As the curing time increased, the size of the free volume holes decreased. The appropriate light-curing polymerization approach depends on the light curing intensity, which is proportional to the depth of the hole in the tooth [110]. PALS was used to evaluate how increasing the degree of co-polymerization of TEGMA influenced free volume formation in Bis–GMA composite systems. The degree of volumetric shrinkage in the system increased with increasing amounts of TEGMA and longer irradiation times for photopolymerization. The relationship between the volume shrinkage and the micro- and macrostructures was more complex. The PALS method was able to track systemic changes in the free volume and network formation related to the polymerization time [111]. The light-curing polymerization process and consequent volumetric shrinkage were examined in the commercially available acrylate-based dental composite ESTA-3^®^ (ESTA Ltd.) using PALS. In the results, Ps was formed in the polymer matrix, and positron trapping occurred in the free volume of the interface between the polymer and the filler. The small degree of shrinkage (1.5%) during photopolymerization can be attributed to the high filler percentage in the composite and the small amount of resin [112]. The photopolymerization shrinkage was examined in commercially available Charisma^®^ dental fillings. By increasing the light exposure of the composites, the o-Ps lifetime values decreased, and this is related to the crosslinking of polymeric chains in the matrix [82]. The free volume and mechanical properties of Palacos^®^ R bone cement were investigated by PALS. The free volume hole size was reduced in the matrix due to the elimination of residual monomers at temperatures higher than 60 °C [114].

##### Conclusions and Discussion of Reviews

The application of PALS as a novel method in the life sciences has been described—specifically, in drug delivery systems, macromolecular and membrane studies, and biological tissue research. PALS provides information at the supramolecular and molecular levels about the free volumes and voids, their distributions within the examined systems, molecular bonds, and the structures at different depths and layers, and is able to track phase transitions. The interpretation of PALS results requires well-characterized, controlled systems, which are not necessarily provided when using biological samples [30,116].

The application of polymeric films in the coating of solid dosage forms was investigated in detail. All amorphous polymers undergo aging during storage, and the microstructural changes at play here can be tracked by PALS. As the polymeric material relaxes, the free volume decreases over time, and densification of the matrix occurs. Temperature, humidity, and the addition of excipients all influence the aging process [44,117,118].

The encapsulation and biostabilization of chemically sensitive bioactive components in glassy carbohydrate matrices consisting of trehalose and diluent glycerol have been widely examined. This review focused on the structural and thermodynamic aspects affecting the performance of the amorphous matrix, studying the plasticization and antiplasticization properties of water and other diluents. The PALS method has revealed that a low diluent content leads to reductions in the molecular hole size. A very narrow antiplasticization regime was observed. The effectiveness of the antiplasticizer in decreasing the hole volume and increasing the matrix density is greater when using lower-molecular-weight diluents than when using high-molecular-weight diluents. The PALS results correlate well with the oxygen uptake and rate of oxidation of encapsulated bioactive compounds. The performances of amorphous matrices, in terms of the encapsulation of sensitive drugs, are related to both, sub-Tg dynamics and molecular packing [119,120]. 

Examinations of the effects of water in a solid state on the physicochemical properties of the system, as well as the drug stability and release during formulation and storage, were performed. At higher concentrations, water acts as a plasticizer in polymeric systems; it increases the distance between polymer chains by forming hydrogen bonds, thus increasing the free volume size. During stability testing, high relative humidity was shown to induce the swelling of polymers, and changes in microvoids can influence the drug release mechanism and dissolution profile [121,122].

PALS is a suitable method for measuring and understanding lot-to-lot variability, as well as critical deviations in physicochemical parameters, between batches of lyophilized materials. PALS is suitable for the detection of free volumes and defects in the matrix, and the results relate to structural relaxation and molecular mobility [123,124].

The theoretical background for using PALS for the analysis of porous systems has been discussed [125].

## 3. Discussion

In the pharmaceutical field, amorphous systems (both in terms of active ingredients and carriers) are preferred over crystalline ones due to the increased solubility and bioavailability attributed to the former. To achieve structural and chemical stability in dosage forms and devices, the precise and careful selection of composition and technology is required, as amorphous systems show high enthalpy and over a certain time period, tend to return to their low-energy crystalline state. The free volumes of the samples primarily analyzed by the PALS method, the relaxation processes occurring over time, and the interactions within the solid matrix directly relate to molecular mobility and can thus be used to determine the release of the active ingredient. 

It is important to note that the samples used for the PALS measurements in the review adopted several forms, such as compacted powders, polymeric films, nanofibers, and commercially available solid products.

PALS, as a microstructural analytical technique, is able to characterize and distinguish drug-free and drug-loaded nanofibrous systems. Drug incorporation in polymeric matrices reduces free volume sizes, which is supported by the “hole-filling” concept. The effects of the applied technological device and its process parameters on fiber morphology and the phase transition of the drug from crystalline to amorphous can also be tracked by PALS. The effects of the composition, the chain length of selected polymers, and possible drug–polymer interactions influence the free volume hole sizes and, consequently, drug mobility in the matrix and its release properties. Furthermore, the aforementioned factors also affect the macromolecular mechanical characteristics.

Amorphous systems undergo physical aging. The aging process can be tracked by accelerated or long-term stability testing under different temperatures and humidity conditions, and PALS is an effective method for monitoring changes in the supramolecular structure during this process. During stability testing focusing on the polymeric aging and relaxation of chains, the free volume usually decreases. Depending on storage conditions, new interactions are formed and destroyed in the matrix, and new structural organizations may develop over time. Systems are considered stable when the microstructure does not change significantly over a long period of time. Microstructural changes and increased molecular mobility during stability testing are attributed to the absorption of moisture, a basic external factor that can induce a glassy to rubbery phase transition and alter the pore structure. The modulation of the porous structure over time affects the diffusion of active substances into and out of the carrier matrix. The water absorbed from the environment plasticizes polymeric systems and increases the sizes of free volume holes by creating H-bonds with the polymeric chains. Depending on the material’s quality and hygroscopicity, a large degree of water absorption can lead to saturation of the system, and swelling can be observed.

Molecular packing, the stability of the incorporated drug and polymers, and the mechanical properties are greatly influenced by the excipients and external stimuli affecting the carrier matrix. The water content and the humidity of the environment around the system during storage can affect the examined system in several ways, depending on the composition of the matrix and other excipients used. At low concentrations, water usually has an antiplasticization effect on the matrix and reduces the free volume size. However, when increasing the water content, the free volume exceeds a minimum value, and water acts as a plasticizer by forming H-bonds with the polymeric chains, thus increasing the free volume of the system. The effect of polyols has been widely examined in amorphous, glassy carbohydrates, which are applied in the biostabilization of sensitive active substances. Low-molecular-weight polyols, e.g., glycerol and sorbitol, enhance molecular packing in certain carbohydrate matrices compared to xylitol, which plasticizes them via the formation of H-bonds. The antiplasticization or plasticization effects of polyols in amorphous carbohydrate systems can be diverse, depending on the material quality of the matrix, as well as the composition and concentration of the polyol and other applied excipients; thus, their effect must be examined in a comprehensive manner. In nanofibrous systems, excipients are used as solubilizing agents to achieve increased aqueous solubility in the drug (polysorbate 80 and polysorbate 20 surfactants). The cyclodextrines used to improve drug stability exert plasticization effects on the polymeric matrix. 

In order to examine the external factors affecting the matrix during the manufacturing process, PVT experiments have been performed, and these highlight the sensitivity of the system, its susceptibility to microstructural changes, and the parameters that must be carefully controlled in order to avoid instability. The occurrence of glass transition upon heating past the required temperature can also be assessed using the PALS method. The thermal expansion of polymeric systems is often observed as a result of heating; differential scanning calorimetry measurements are recommended to correlate and interpret the results. Some modern drug delivery systems have temperature-responsive drug release characteristics related to the modulation of microstructural properties. Increasing the temperature induces an increase in the free volume size and fraction and, thus, the diffusion of the drug from the carrier matrix. The PALS method is suitable for monitoring this change during the development of the drug product.

Defect characterization is a widely analyzed factor in the field of contact lens materials (rigid gas-permeable lenses, soft hydrogel lenses, and silicone-hydrogel lenses). The geometry and nature of defects and vacancies vary greatly, and this is attributed to the different preparation technologies used in developing the materials. Furthermore, the free volume size and fraction determine the oxygen permeability and diffusion of water and ions, which may compromise the healthy pathological conditions of the eye as well as patient comfort.

PALS is not only suitable for the examination of complex drug delivery systems and delivery bases but can also be used to investigate the microstructural and structural differences of active substances and supramolecular structures, such as drug complexes, organic compounds, and dendrimers. The absence and presence of functional groups and free voids are revealed by spectral parameters. 

### 3.1. Validation of the Results of the PALS Method

As with any analytical method, PALS needs to be validated via an independent method to verify that the measurements (lifetime) and calculations of the results (free volume) are correct. In the case of PALS, there are several techniques available in the literature to do this. The most important techniques are listed below.

### 3.2. Brunauer, Emmett, and Teller (BET) Method 

This technique involves the formation of a layer of liquid nitrogen (at 77 K) on the sample’s surface to obtain information on the material’s specific surface area and porosity. The BET method is less suitable for the analysis of micropores, as N_2_ has a diameter slightly greater than that of the Ps atom (1.06 Ǻ), meaning the latter cannot penetrate closed pores. The two methods applied to medium-sized pores yielded identical results, as the free volume (ΔR) constant was derived from BET measurements [126].

### 3.3. 129-Xe Magnetic Resonance Spectroscopy (Xe-NMR)

The measurement of cavity dimensions using 129-Xe NMR is based on similar considerations as the BET measurement. An NMR tube containing a sample is filled with xenon gas at a pressure of about 5 atm. The Xe diffuses into the cavities, and its measured chemical shift (in ppm, as used in NMR, relative to free Xe gas) is taken as proportional to the reciprocal of the detected cavity size. The disadvantage of this measurement method is that it has to be corrected for the chemical shift of Xe absorbed in the NMR tube wall; the Xe atom has a diameter of 4.4 Ǻ, so smaller cavities are not detected, and it does not enter closed cavities as N_2_ does. The costs of the maintenance of the NMR instrument, mainly due to the cooling of the superconducting magnet with liquid helium, are much higher than those for PALS [127]. 

#### 3.3.1. Differential Scanning Calorimetry (DSC)

DSC is a well-known thermoanalytical method in which the temperature of the sample being tested varies in a closed system, while the amount of heat required to induce the temperature change is measured [128]. If a structural change occurs in the sample, the slope of the hitherto nearly linear curve (which becomes non-linear because the specific heat varies with temperature) changes; depending on whether the change is exothermic or endothermic, we see a decrease or an increase. For polymers, the T_g_ is most often determined using DSC, while temperature-dependent PALS or DBES (Doppler Broadening Energy Spectra) can also measure this parameter [129], demonstrating that the varying measurement results indicate a structural change. For some polymer systems, the DSC curve is difficult to evaluate, and positron measurements can help determine the Tg. However, the routine use of this approach is hampered by the speed of performing DSC about 120 times (typical heating rates: DSC: 10 K/min, PALS: 5 K/h [130]. 

#### 3.3.2. Imaging Techniques

For the detection of the microstructure, which is analyzed using positron techniques, the use of a microscope seems the obvious choice. Of course, such small details of the material cannot be seen with a conventional microscope, but microscopic techniques offer sufficient magnification to examine tiny cavities. Two of the most important techniques are atomic force microscopy (AFM) and scanning electron microscopy (SEM). AFM images at the same scale as PALS, but SEM only images at approximately 10 Ǻ or larger, so although an SEM photo series can be used to form a PALS measurement, the changes seen and measured are only correlated, not identical. Both methods (AFM and SEM) provide images of the surface of the material under investigation, unlike PALS, which measures the top 100 µm of the material, but while AFM is three-dimensional, SEM shows only a 2D projection. A further difference is that AFM does not require special sample preparation, nor does PALS, and no vacuum is needed. The advantage of the SEM, on the other hand, is its speed; although it has a lower resolution than the AFM, it can show almost “live” images of several times the maximum scanning area of the AFM (150 µm x 150 µm).

#### 3.3.3. Complementary Use of PALS with Other Analytical Methods

The complementary use of analytical methods for the physical and chemical characterization of solid-state pharmaceutical systems forms a complex picture of the examined matrix and the effects of excipients and external factors. In this chapter, the methods that are typically applied together are discussed based on the scientific literature; the conclusions apply to the pharmaceutical systems and conditions described in this area.

SEM is a common imaging technique used for the visualization of microscopic and surface topography and generates high-resolution images of samples from the nm to µm scale. According to the nanofibrous matrix morphological analysis, the orientation and homogeneity of fibers are directly connected with the supramolecular organization of the polymeric chains. The observed phenomena are supported by possible interactions between polymers and the active substance, the excipient. The images also help us to determine the average fiber diameter of the web and track morphological changes related to the modification of technological parameters, external stimuli, or stability testing. 

Among the macrostructural characterization techniques, DSC is a thermoanalytical method applied for the determination of the T_g_, which is a parameter closely related to the free volume change. At *Tg*, the system undergoes glassy-to-rubbery phase transition, and this structural change is often inferred using the free volume concept. Materials with higher *Tg* values undergo smaller expansions upon temperature modification. Apart from the systemic temperature dependence, the chemical interactions formed in the matrix are also of key importance when evaluating analytical results. FTIR and ATR-FTIR provide information about the physicochemical properties of the system and possible chemical interactions between its components, i.e., the carrier polymer, the active substance, and the excipients. During the preparation of nanofibers via the electrospinning process, the solid-state phase transition from crystalline to amorphous is often tracked using the FTIR method. FTIR complements PALS measurement; as a solid-state characterization technique, the vibrations and characteristic peaks in the FTIR spectra (associated with functional groups at specific wavelengths) provide information about the quality of the examined material. The XRD method is applied to examine the crystallinity of the samples. In the literature, it is mainly applied to test whether the initially crystalline active substance completely transforms into its amorphous equivalent during the electrospinning process. RS is based on the vibrational and rotational characterization of molecules, which is attributed to the molecular mobility of moieties and conformations in the molecular structure.

ssNMR, similarly to PALS, is a microstructural characterization method, but it is based on the nuclear spin interactions of systems containing magnetic nuclei. Primarily, information related to quality is obtained, but it also provides quantitative information.

## 4. Research Methods 

The literature search, as well as the collection, screening, and selection of scientific articles related to the PALS method of solid-state characterization, was performed according to the guidelines of the Preferred Reporting Items for Systematic Reviews and Meta-Analyses (PRISMA) 2020 model.

### 4.1. Eligibility Criteria

Scientific articles were included for data processing if they met the three eligibility criteria: the application of the PALS method, a focus on a field of application in the pharmaceutical or medical industry, and the examination of a solid pharmaceutical dosage form or medical device. The systematic review covers drug delivery systems and devices without active ingredients, such as contact lenses, intraocular lenses, and dental implants; furthermore, it examines the effects of excipients and external factors on the aforementioned systems. Regarding the classification of articles, only peer-reviewed original articles and reviews are included; conference abstracts, patents, and case studies are excluded. In addition to the utilization of the PALS method, a strong emphasis had to be placed on the comparison of the method with other structural analysis methods to determine the advantages, disadvantages, and possibilities of parallel use to extract information about combinations of methods.

### 4.2. Search Strategy

The systematic review included scientific publications from the last 20 years (2002–2022). Four databases were searched: PubMed, Embase, SciFinder-n, and Google Scholar. An additional condition of the search strategy was that only articles in English were selected for further screening. At the beginning of the search, the basic search term was “Positron annihilation lifetime spectroscopy” (PALS); the results included all subject areas and the examination of solid as well as non-solid systems. In a subsequent step, in order to reduce the set of results, additional search terms were used, specifically: “(PALS AND (drug)”, “(PALS) AND (excipient)”, “(PALS) AND (dosage form)”, “(PALS) AND (medicine)”, “(PALS) AND (pharmaceutics)”, and “(PALS) AND (polymer)”. The results were then collected in tabular form according to databases.

### 4.3. Data Analysis

Based on the results of the database matches, scientific articles with titles that fulfilled the criteria described previously or whose topics were not clearly revealed by their titles were included for further pre-screening. The articles and reviews that met the eligibility criteria based on their abstracts were collected. According to the article title or abstract, if the topic of discussion was not related to solid systems or pharmaceutics and medicine, the article was excluded. The EndNote reference manager tool was used for the easy and efficient management of article links. Duplicates were removed, and the selected articles were processed using a predetermined logical outline based on the following steps:(a)Dosage form and drug;(b)Carrier system;(c)PALS as a basic or supportive testing method, in addition to other microstructural methods;(d)The aim of the study;(e)Results;(f)Conclusion.

## 5. Conclusions

This review highlights the extensive and effective application of the PALS method in a particular sub-domain of the material sciences, i.e., the pharmaceutical sector, for the microstructural analysis of solid systems, and more precisely, active pharmaceutical ingredients, excipients, drug delivery bases, drug delivery systems, and medical devices. 

With this technique, it is possible to examine the free volume hole sizes and fractions and the variability of these parameters over time, as well as to detect the effects of external factors. From a spectroscopic point of view, one of the most important properties of PALS is that when applied to o-Ps, the free volumes in materials (cavities, vacancy, vacancy groups, and holes) appear as potential wells. The Ps is typically localized in such free volume holes, and the local electron density of the hole determines its lifetime. This is the unique advantage of lifetime spectroscopy over conventional structural methods, which give a more accurate picture of the molecular structure but fail to penetrate the free volumes [131]. 

In terms of method functionality, the microstructural features of the systems can be revealed and tracked; moreover, the method is able to reveal defects and voids and characterize nanostructural components. PALS is of essential interest in polymeric systems, particularly amorphous or partially amorphous polymers. In amorphous polymers, free-volume holes are not uniformly distributed. Instead of a well-defined o-Ps state, many similar states arise depending on the sizes of the free volume holes around the Ps. Since pharmaceutical dosages comprise multicomponent systems, primarily built out of polymeric excipients and active substances, changes in their supramolecular properties over the course of formulation and storage can be sensitively tracked using the PALS method, thus predicting possible destabilizing interactions. In addition, this method can be used for testing and identifying the extent of defects derived from the manufacturing technology used in polymer-based medical devices (mainly contact lenses).

However, using the method alone is not suitable for determining the nature of chemical bonds and functional groups, but with the addition of other analytical methods (DSC, FTIR, XRD, RS, and ssNMR), it provides a comprehensive physicochemical profile of the system. The application of this method in pharmaceutical technology could be useful for microstructural monitoring, the identification of material and functional relationships, stability prediction, and delivery systems’ drug storage capacity estimations depending on free volume sizes and distributions. 

PALS enables the tracking of active substances, excipients, dosage forms, and medical devices from the point of their supramolecular changes, as summarized in the review. By comparing the measurement results with the mechanical properties, phase transformations, and dissolution profiles associated with treatment and storage, we can see how the supramolecular structures of the materials used, as yielded by PALS, are related to many measured and measurable material properties of pharmaceutical importance.

## Figures and Tables

**Figure 1 pharmaceuticals-16-00252-f001:**
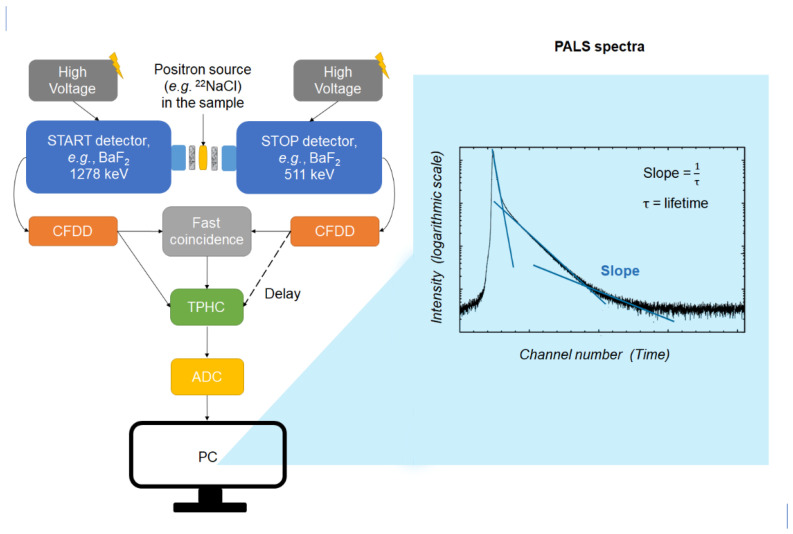
Schematic of the PALS spectrometer and spectra. Abbreviations: ADC = analogue-to-digital converter, CFDD = constant-fraction differential discriminator, TPHC = time-to-pulse height converter, PC = personal computer and special program for experimental data collection and analysis. Notes: The fast coincidence circuit produces the gate signal for the TPHC provided that the coincidence event of gamma rays with proper energies has occurred simultaneously.

**Figure 2 pharmaceuticals-16-00252-f002:**
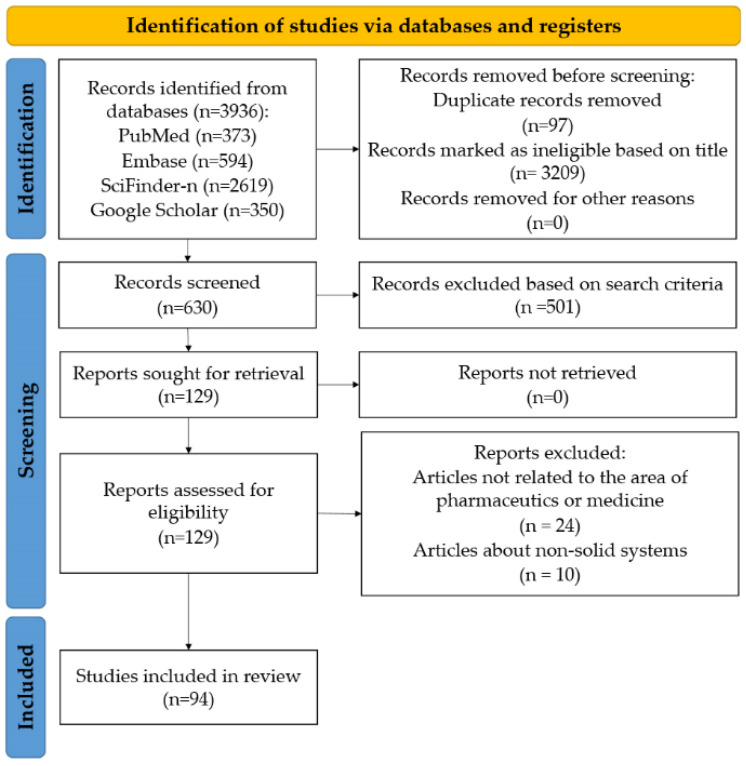
Steps of the selection and screening of the scientific publications according to the PRISMA 2020 flow diagram.

**Table 1 pharmaceuticals-16-00252-t001:** Positron annihilation characteristics.

Positron Annihilation Mode	Characteristic Lifetime in Vacuum	Emitted Photons in Vacuum	Positron Annihilation in Matter	Ref.
p-Ps ^1^ (with antiparallel orientation of the positron and electron spins)	0.125 ns	Two 0.511 MeV gamma rays	The p-Ps lifetime can be affected by the material because the Coulomb interaction between the p-Ps and material electrons changes the distance between the positron and electron in p-Ps. However, the p-Ps lifetime remains relatively unchanged by the interaction with condensed matter as it undergoes self-annihilation.	[1,3,19]
o-Ps ^2^ (with parallel orientation of the positron and electron spins)	142 ns	Three 0.511 MeV gamma rays constrained by the conservation of angular momentum	Self-annihilation is forbidden by quantum mechanics in the triplet state of o-Ps, and its lifetime in a material is drastically reduced. o-Ps mainly decays via the “pick-off” process where the positron is annihilated along with an electron with opposed spin in the surrounding material; two 0.511 MeV annihilation photons are created, and the lifetimes are shortened. These lifetimes are still longer than the mean lifetime of p-Ps and also long enough for Ps atoms to “scan” their surroundings and be easily measured.	

^1^ p-Ps = para-positronium, ^2^ o-Ps = ortho-positronium.

**Table 2 pharmaceuticals-16-00252-t002:** Summary of scientific publications related to the PALS method.

Dosage Form	Drug and Concentration	Carrier Matrix (Chemical and Commercial Name)	Aim of Study	Functionality of the PALS Method	Other Additional Structural Analyses Used Besides PALS ^1^	References
(1) Substances						
(a) Active pharmaceutical ingredients						
Amorphous drug film	Verapamil hydrochloride	-	Characterization of the temperature dependence of the free volume	Microstructural analysis	DSC BDS	[35]
Powder	Metformin drug complex with vanadium(III) and chromium(IV) ions	-	Examination and physicochemical characterization of the system	Microstructural analysis	SEM IRS RS	[36]
Powder	N-heterocyclic compounds	-	Investigation of the structural characteristics of the system	Microstructural analysis	DBES	[37]
Crystalline powder	Olanzapine	-	Examination of temperature and pressure dependence of the drug microstructure	Tracking of microstructural changes upon external stimuli (heat and pressure)	PVT	[38]
(b) Excipients						
Powder	-	Poly(ethylene oxide) (PEO)	Investigation of poly(ethylene oxide) ageing	Microstructural analysis Tracking of microstructural changes during accelerated stability testing (40 °C, 75% RH, 4 weeks)	SEM DSC	[39]
(c) Processed excipients						
Pellet cores	-	Microcrystalline cellulose and isomalt	Examination of the relationship between physical attributes and the supramolecular structure of the pellet cores	Microstructural analysis	SM SEM ATR-FTIR NIR	[40]
Pellets	-	Polyvinylpirrolidone (PVP)	Investigation of PVP as a stabilizer and binder by the PALS method	Microstructural analysis	XRPD RS SEM EDS	[41]
Pellets (melt extrudate)	Mebendazole (20 g/50 g formulation)	Soluplus, Eudragit E PO, povidone (Kollidon K30, PVP30), and Plasdone C17	Examination and physicochemical characterization of the system	Microstructural analysis	SEM DSC FTIR	[42]
Spray-dried polymeric blends	-	Hydrophobically modified starch and sucrose	Investigation of the structure and the effect of sucrose and water in the starch–sucrose phase-separated system	Microstructural analysis	ssNMR	[43]
(2) Delivery bases and drug delivery systems						
(a) Free and drug-loaded polymer films						
Polymeric film	-	Polyvinylpirrolidone (PVP) and poly(ethylene glycol) (PEG), PVP-PEG diacrylate (PVP-PEGDA), PVP-PEG monomethacrylate (PVP-PEGMMA)	Examination of the effect of hydrogen bonding on the depth profile of the free volume in polymeric mixtures and copolymers	Microstructural analysis	DBES	[44]
Polymeric film	-	Sodium alginate (SA), Carbopol 71 G	Tracking of the microstructure in real time during film formation	Microstructural analysis	-	[45]
Polymeric film	-	Ethylcellulose and PVP blend	Studying phase separation in the polymeric blend	Microstructural analysis	SEM ATR-FTIR EDX DSC	[46]
Buccal film	-	Hydroxypropyl-methylcellulose-5 (HPMC-5), Hydroxypropyl-methylcellulose-15 (HPMC-15)	Examination of the effect of the sucrose palmitate permeation enhancer on the physicochemical and mucoadhesive properties of the system	Microstructural analysis	XRD	[47]
Solvent cast polymeric film	-	Acrylic polymers (Eudragit L 30D and Eudragit RL 30D)	Examination of the plasticizer effect of dibutyl sebecate on the system	Microstructural analysis	-	[48]
Polymeric film	-	Bovine gelatine	Studying the effect of polyol plasticizers on the enthalpy of the system	Microstructural analysis	DSC	[49]
Solvent cast polymeric film	-	Maltopolymer–maltose blends	Studying the molecular packing and the effect of water in amorphous carbohydrate matrices	Microstructural analysis	Density measurement	[50]
Solvent cast polymeric films	-	Maltodextrin and dextrose	Investigation of the structure of amorphous carbohydrate matrices by combined methods	Microstructural analysis	DSC Density measurement Dilatometry	[51]
Solvent cast polymeric film	-	Maltopolymer–maltose blends	Examination of the different effects (temperature, pressure, and water content) on the system	Microstructural analysis	PVT	[52]
Polymeric film	-	PVP, Eudragit NE 30 D	Examination of the free volume of polymers under different humidity conditions	Microstructural analysis Tracking of microstructural changes during accelerated stability testing (25 °C; 45, 55, 65, 75% RH, 30 days)	-	[2]
Polymeric film	-	Methylcellulose	Investigation of the effect of PEG plasticizer on the aging of the system	Microstructural analysis Tracking of microstructural changes during accelerated stability testing (25 °C; 75% RH, 30 days)	DBES	[53]
Solvent cast polymeric film	-	Methylcellulose	Investigation of PEG-Metolose interaction and aging of the system by the PALS method	Microstructural analysis Tracking of microstructural changes during accelerated stability testing (25 °C; 75% RH, 30 days)	DSC DBES	[54]
Polymeric film	-	Gelatine and sucrose blend	Examination of the effect of water and drying on the system	Tracking of microstructural changes upon external stimulation (moisture)	-	[55]
Polymeric film	Bicalutamide	PVP	Investigation of the effect of the polymer chain length on the recrystallization of amorphous drugs	Microstructural analysis	DSC BDS	[56]
Spray-dried dispersions and polymeric films	Indomethacin Ketoconazole	PVP, Polyvinylpirrolidone-vinylacetate (PVP-VA), Hydroxypropylcellulose (HPC), Hydroxymethylcellulose acetate succinate (HPMC-AS)	Investigation of the effect of excipients on the structure of amorphous dispersions	Microstructural analysis	SEM PXRD DSC DMA PVT	[57]
Mucoadhesive polymeric film	Lidocaine base (5, 10, 15 w/wt. %)	HPC	Examination of the plasticizer effect of polyols (glycerol and xylitol) on the system	Microstructural analysis	-	[58]
Polymeric film	Penicillin G	Polyethylene (PE) and polypropylene (PP)	Examination and characterization of the system	Microstructural analysis	FTIR SEM XPS	[59]
Nail lacquers	Terbinafine HCl (1.0% *w*/*w*)	Polyurethane (PU)	Development of the system, microstructural analysis, biocompatibility, wettability, and antifungal activity testing	Microstructural analysis	SEM FTIR	[60]
Solvent cast and freeze-dried polymeric films	Vitamin B_12_ (2 mg Vit B_12_/10 g)	SA and Carbopol 71G	Finding correlations between the drug release profile and supramolecular structure	Microstructural analysis Tracking of microstructural changes during accelerated stability testing (40 °C, 75% RH, 4 weeks)	Digital microscope	[61]
Polymeric film	Diclofenac sodium (0, 1, 5 w/wt. %)	Eudragit L 30D-55	Investigation of the structure of the system with variable drug content and during storage	Microstructural analysis Tracking of microstructural changes during accelerated stability testing (17 °C; 65% RH, 3 weeks)	-	[62]
Spray-dried dispersions	Indomethacin Ketoconazole	PVP, PVP-VA copolymer, HPC, HPMC	Examination of drug recrystallization propensity in different carrier systems, moisture sorption, and relaxation	Microstructural analysis Tracking of microstructural changes upon external stimulation (moisture)	Polarized microscopy XRPD	[63]
Polymer membrane	Sulfamethoxazole (3.5 wt. %), Paclitaxel (3.0 wt. %)	Polyurethane	Investigation of the structure and drug release of the temperature-modulated drug delivery system	Tracking of microstructural changes upon external stimulation (heat)	DSC	[64]
(b) Neat and loaded micro- and nanofibers						
Buccal nanofibrous sheets	Nebivolol hydrochloride (0.1 g/20 g water)	PVA	Development of the system for enhanced dissolution of BCS II drugs	Microstructural analysis	ATR-FTIR DSC	[65]
Topical nanofibrous sheets	Iodine (2.7–3.1 w/wt. %)	PVP, PVP-VA copolymer	Preparation of the system, physico-chemical characterization	Microstructural analysis	SEM	[66]
Buccal nanofibrous sheets	Papaverine hydrochloride (30 mg/g stock solution)	HPC and poly(vinyl alcohol) (PVA) composite	Examination of the microstructure of gels, films, and nanofibers	Microstructural analysis	ssNMR	[67]
Rotary spun microfibers—tablets	Vitamin B_12_ (5 mg/mL)	PVP	Investigation of the structural and mechanical properties of the system	Microstructural analysis	SEM Density measurement	[68]
Micro- and nanofibers	-	PVP, PVP-VA copolymer	Studying the influence of parameters for optimum fiber morphology and the mechanical properties of the system	Microstructural analysis	SEM	[69]
Nanofibrous sheet	-	PVP, PVP-VA copolymer	Development of a high-speed rotary jet device and examination of the correlation between preparation parameters and fiber morphology	Microstructural analysis	Optical microscopy SEM	[70]
Buccal nanofibrous sheets	Fenofibrate (0.2 g fenofibrate/5 mL solution)	PVP	Development and physicochemical characterization of the system	Microstructural analysis	SEM ATR-FTIR	[71]
Topical nanofibrous sheets	Colistin-sulfate (15 w/wt. %)	PVA	Preparation of complex, reservoir-type nanofibrous wound dressings	Microstructural analysis Tracking of microstructural changes upon external stimuli (heat)	-	[72]
Rotary spun microfibers—orodispersible tablets	Carvedilol (5g/50 mL solution)	HPC	Formulation of the system and tracking the crystalline–amorphous transition of the drug during preparation and stability testing	Microstructural analysis Tracking of the microstructure over accelerated stability testing (40 °C, 75% RH, 4 weeks)	XRPD DSC ATR-FTIR	[73]
Buccal nanofibrous sheets	Papaverine hydrochloride (30 mg/g stock solution)	HPC and PVA composite	Monitoring of supramolecular changes of the system under stress conditions	Microstructural analysis Tracking of polymer aging over accelerated stability testing (40 °C, 75% RH, 4 weeks)	SEM FTIR RS	[74]
Buccal nanofibrous sheets	Furosemide (1 w/wt. %)	PVP, HPC	Examination of different solubility-enhancing excipients in the prepared systems	Microstructural analysis Tracking of microstructural changes during accelerated stability testing (40 °C, 75% RH, 4 weeks)	SEM ATR-FTIR XRD	[75]
Buccal nanofibrous webs	Metoclopramide hydrochloride (3 w/wt. %)	PVA	Examination of the effects of Polysorbate 80 and hydroxypropyl-β-cyclodextrin on the electrospinning process and mechanical properties of the system	Microstructural analysis Tracking of microstructural changes during accelerated stability testing (40 °C, 75% RH, 4 weeks)	SEM AFM PXRD NMR ssNMR	[76]
3D-printed drug delivery base						
Polyvalent test plates (PVTP)	-	Oligoamine-modified poly(lactic acid)	Investigation of the connection between the chemical and structural parameters and the cytocompatibility of the chemically modified system	Microstructural analysis	Optical microscopy SEM FTIR	[77]
*Living organism-loaded system*						
Nanofibrous web	*Stenotrophomas maltophilia*	PVA	Tracking of bacterial viability in bacteria-embedded webs	Microstructural analysis Tracking of microstructural changes upon external stimuli (bacterial metabolic product) Tracking bacterial viability over accelerated stability testing (40 °C, 75% RH, 3 weeks)	SEM ATR-FTIR	[78]
(c) Freeze-dried formulations and biologics						
Lyophilized powder	-	Sodium hyaluronate	Studying the structural and mechanical properties of the system	Microstructural analysis	SEM XRPD	[79]
Freeze-dried matrices	-	Maltodextrin	Examination of the effect of glycerol and water excipients on the hydrogen bonding within the system	Microstructural analysis	FTIR DSC	[80]
Freeze-dried powder	-	Maltodextrin	Studying the effect of water and glycerol on the structural properties of the system	Microstructural analysis	DSC	[81]
Freeze-dried powder	-	Sucrose and starch	Examination of the effect of temperature on the free volume of the system	Microstructural analysis Tracking of microstructural changes upon external stimulation (heat and pressure)	TGA	[82]
Lyophilized formulations	Human growth hormone (hGH)	Hydroxyethylstarch (HES)/disaccharide	Evaluation of the effect of disaccharide and polyols on the free volume changes of the system	Microstructural analysis	He pycnometry NS	[83]
Freeze-dried formulation	Bovine serum albumin (BSA) and recombinant human serum albumin (rHSA)	Sucrose as a base material and alkali halides (LiCl, NaCl, KCl, RbCl, CsCl) as excipients	Evaluation of the effect on low-level electrolytes on the stability of formulations under stress conditions	Tracking of the microstructure over accelerated stability testing (50 °C, 6 weeks for rHSA and 50 °C, 65 °C, 2 months for BSA samples)	FTIR TAM NS	[84]
(d) Tablets						
Tablet	Theophylline (100 g/10 mL solution)	PVP	Examination of the drug release properties and free volume of the system under different humidity conditions	Microstructural analysis Tracking of microstructural changes during accelerated stability testing (25 °C; 35, 45, 55, 65, 75% RH, 30 days)	-	[85]
Tablet	Famotidine (30 mg/181 mg formulation)	PVP and Carbopol matrix	Investigation of the effect of water on the structure and drug release of the system during storage	Microstructural analysis Tracking of microstructural changes during accelerated stability testing (40 °C, 75% RH, 4 weeks)	-	[86]
(e) Transdermal patches						
Transdermal patch	Metoprolol tartarate (3 w/wt. %)	Hypromellose and methylcellulose	Examination of drug release	Microstructural analysis	-	[87]
Transdermal patch	Zolmitriptan	Pirrolydone adhesive Amide adhesive	Examination of controlled drug release from acrylate adhesives focusing on the role of hydrogen donors	Microstructural analysis	FTIR DSC	[88]
Transdermal patch	Metoprolol tartarate	HPMC and Eudragit NE 30D	Investigation of the free volume and drug release from cellulose and acrylate type polymers	Microstructural analysis	-	[89]
(f) Intrauterine delivery systems						
Intrauterine system	Levonorgestrel (52 mg/intrauterine device)	Polydimethylsiloxane (PDMS)	Tracking the morphology, microstructural changes, and long-term stability of the system	Microstructural analysis Tracking of microstructure over long-term stability testing (in vivo, 5 years)	SEM	[90]
(g) Nanostructured delivery bases and drug delivery systems						
Dendrimer	-	Polyphenylene	Study of the free volumes of rigid polyphenylene dendrimers	Microstructural analysis	-	[91]
Nanoceria	Cerium dioxide	Microcrystalline cellulose	Synthesis of a hybrid nanostructure and structural and spectroscopic characterization of the system	Microstructural analysis Defect characterization	XRD FEG-SEM HR-TEM FTIR RS	[92]
Nanocomposite	ZnO	Chitosan	Development and fine-tuning of antibiocidal ZnO/chitosan nanocomposite	Microstructural analysis Defect characterization	XRD HR-TEM FTIR	[93]
*Nanostructured sensors*						
Nuclear sensors (radiotracers)	Cu^2+^ and Co^2+^ polyazacarboxylate macrocycles, hexa-aza cages	Hollow silica shells	Determination of the binding properties of porous materials and screening by radiotracers	Microstructural analysis	-	[94]
(3) Medical devices						
(a) Contact lenses and intraocular lenses						
Contact lenses	-	Hydrogel, silicone-hydrogel	Determination and comparison of free volumes and the water content in lenses with different materials	Microstructural analysis	MIR RS	[95]
Contact lenses	-	Hydrogel, silicone-hydrogel, fluorosilicon-methacrylate-copolymer	Comparison of free volume gaps in contact lenses of different polymer types	Microstructural analysis	-	[96]
Contact lenses	-	Hydrogel, silicone-hydrogel	Investigation of the void size and free volumes in the system	Microstructural analysis	-	[97]
Contact lenses	-	Hydrogel, silicone-hydrogel	Investigation of the microstructure in different contact lens materials	Microstructural analysis	-	[98]
Contact lenses	-	Hydrogel	Examination of the effect of the preparation technology and degree of defect in the structure	Microstructural analysis Defect characterization	UV-vis-NIR	[99]
Contact lenses	-	Hydrogel, silicone-hydrogel	Studying the structure of contact lenses and their effect on oxygen permeability	Microstructural analysis Defect characterization	-	[100]
Contact lenses	-	Hydrogel, silicone-hydrogel	Comparison of the degree of disorder of lenses with two-state model and Tao–Eldrup model	Microstructural analysis	-	[101]
Contact lenses	-	Poly (fluorosilicone acrylate)	Investigation of water and glucose diffusion through the system	Microstructural analysis	-	[102]
Contact lenses	-	Poly(2-hydroxy ethyl methacrylate) (PHEMA)	Examination of calcification in soft contact lenses and their effect on free volume holes and optical properties	Microstructural analysis	EDS	[103]
Intraocular lenses	-	Polymeric combinations of methyl methacrylate (MMA), buthyl methacrylate (BMA), ethy hexyl acrylate (EHA) monomers	Examination of the free volume in vinyl polymer-based intraocular lenses with different compositions	Microstructural analysis	-	[104]
Intraocular lenses	-	2-phenylethyl acrylate (PEA) and 2-phenylethyl methacrylate (PEMA) copolymer	Examination and comparison of the structure of intraocular implants	Microstructural analysis	-	[105]
Intraocular lenses	-	Hydroxyethyl-2-metacrylate (HEMA)	Examination of the time-dependent impact of silicone oil on the system	Microstructural analysis Tracking of microstructural changes during long-term stability testing (37 °C, 6 months)	ATR-FTIR	[106]
Intraocular lenses	-	Poly(methyl methacrylate) (PMMA), PHEMA	Investigation of the effect of silicone oil on the internal structure of intraocular lenses	Microstructural analysis	FTIR RS	[107]
(b) Dental fillings						
Dental filling	-	Bisphenol A-Glycidyl Methacrylate (Bis-GMA) and Tri-ethylene glycol dimethacrylate (TEGMA)	Studying the influence of aging on the dental polymer material	Microstructural analysis Tracking of microstructural changes during long-term stability testing (2 years)	-	[108]
Dental filling	-	Bis-GMA and TEGMA	Examination of photopolymerized dimethacrylate-based dental restorative composites	Microstructural analysis	-	[109]
Dental filling	-	Coltene, Filtek Z250, DenFil™, Heliomolar^2^	Comparison of the photosensitivity and free volumes of four dental restorative materials	Microstructural analysis	-	[110]
Dental filling	-	TEGMA and Bis-GMA	Characterization of structural properties in crosslinked dimethacrylate dental composites	Microstructural analysis	NIR	[111]
Dental filling	-	ESTA-3^® 2^	Examination of volumetric shrinkage during the light-curing polymerization process of the system	Microstructural analysis	-	[112]
Dental filling	-	Charisma^® 2^	Studying the photopolymerization shrinkage in the system	Microstructural analysis	-	[113]
Bone cement	-	Palacos R^®^ bone cement^2^	Investigation of the effect of residual monomers on the free volume and mechanical properties of the system	Microstructural analysis	DMA	[114]

^1^ Abbrevations: sp. = spectroscopy; DSC = Differential Scanning Calorimetry; BDS = Broad-band Dielectric Spectroscopy; SEM = Scanning Electron Microscopy; IRS = infrared spectroscopy; RS = Raman spectroscopy; DBES = Doppler Broadening Energy Spectroscopy; PVT = Pressure–volume–temperature measurement; SM = stereomicroscopy; ATR-FTIR = Attenuated Total Reflectance Fourier-Transform Infrared Spectroscopy; NIR = near-infrared spectroscopy; XRPD = X-ray Powder Diffraction; EDS = Energy Dispersive X-ray Spectroscopy; ssNMR = solid-state Nuclear Magnetic Resonance; FTIR = Fourier-Transform Infrared Spectroscopy; EDX = Energy Dispersive X-ray analysis; XRD = X-ray Diffraction; DMA = dynamic mechanical thermal analysis; AFM = Atomic Force Microscopy; NMR = Nuclear Magnetic Resonance; TGA = Thermal Gravimetry Analysis; TAM = Thermal Activity Monitor, refers to isothermal calorimetry; NS = Neutron Scattering; FEG-SEM = Field Emission Gun Scanning Electron Microscopy; HR-TEM = High-resolution Transmission Electron Microscopy; MIR = middle-infrared spectroscopy, Calorimetry. ^2^ Commercially available dental composites.

## Data Availability

Not applicable.

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
