# Peer review of "Positron Annihilation Lifetime Spectroscopy as a Special Technique for the Solid-State Characterization of Pharmaceutical Excipients, Drug Delivery Systems, and Medical Devices—A Systematic Review"

_pharmaceuticals, 2023, doi:10.3390/ph16020252_

Round 1

Reviewer 1 Report

The novel experimental methods of positron annihilation lifetime spectroscopy have been carried out to investigate the microstructural and defects in solid state materials as it's ability to characterize the distribution of electrons aroud atom and vacancy those affected by the enviroment of trapping sites.

For the pharmaceutical excipitents, drug delivery systems and devices, this methods was performed as a usefl tools at many fragmented research papers. The idea of this paper is very interested to collect the scientific literature of the last 20 years and analyize these research results trying to get a review blue map of the application of PAS in this research filed.

But the deficiencies of this paper did not  give the  route map or applicable new methods or technology of PAS during performed to medical and pharmaceutical industry.

Author Response

Dear Reviewer,

Thank you for the valuable comments, which provided a profound initiative for improving our paper. We accepted all the comments and modified the paper accordingly.

Yours sincerely,

Romána Zelkó

Reviewer 2 Report

Reviewer report of the manuscript entitled: Positron annihilation lifetime spectroscopy as a special mean in  the  solid-state  characterization  of  pharmaceutical  excipients, drug delivery systems and devices, by M. Majida et al.

The author performed the review of the literature where the PALS technique was applied to the studies of pharmaceutical, drugs and other chemical compound in medicine.

The text as it is cannot be published because of several mistakes, see below.

In the titled the authors should and the word “Review”, to inform that they do not present their own results.

This sentence:

The annihilation of positrons in collision with electrons is accompanied by the emission of at least two photons (0.511 MeV gamma rays) to conserve energy” is not correct. The fact is that a collision does not mean annihilation. Both processes are different. The positron and the electron can collide but cannot annihilate. Annihilation is a quantum electrodynamic process.

Thermalized positron with an electron of the medium can also form a quasistationary state  called  positronium  (Ps),  a  “hydrogen-like  atom”. The Ps state can occur when a condition other than thermalization is met. The medium must have enough space in the microstructure to produce this condition.

In vacuum 124 ns, in, This is not true.

Three gamma rays with energy lower than   0.511 MeV.  That's not true. Gamma is formed in nuclear processes. Annihilation photons in the 3-photon process have a continuous energy spectrum with an endpoint at 511 keV.

Other methods for the observation of positron annihilation in matter measure the  deviation from collinearity of the annihilation photons or the line shape parameter of the  Doppler-broadened annihilation photons.  Currently, no one uses the annihilation ray angular correlation technique for positron studies. This is the past.

Figure 1 was made using only Word graphics and looks like it was drawn by a child. First, many researchers now use a digital PALS spectrometer, the one pictured is an old analog spectrometer. The authors don't take PALS measurements, that's clear to me, otherwise, they would have presented an actual PALS spectrum instead of a manual one. The authors should ask colleagues, even from the internet, for a real PALS spectrum only for demonstration. I think it would not be a problem to receive it from a lab where PALS is done.

Figure 3 is also very naive.

The authors use only one citation Ref. [1] to the several important achievements in positron annihilation. For instance, Tao-Eldrup equation has the citation [1], this is not fair and correct. The authors should not take shortcuts.

The authors review about 80 papers and Table 2 summarizes the results of investigations paying attention to the results of the PALS method. Looking into the contents of this Table one can conclude that PALS provides information mainly about “microstructure analysis”.  I am not sure if such information is relevant for drugs, medicine, and their application in medicine. Taking into account that the number of publications about PALS is increasing year by year, the information gathered in the paper can be quickly old and not useful. From that point, I doubt if the work of the authors will be useful for readers. Nowadays, via Internet, there is no problem reaching each paper in seconds.

The review is focused on the period from 2003 to 2022 only. Is it any reason to choose this one? PALS are used for over 80 years. Nobody before 2003 has published on this subject

Frankly, I doubt the usefulness of PALS in medicine. The annihilation process is running in the subatomic region whereas the biological processes run at the molecular scale. The authors did not convince me, because each process described by the authors can be easily observed by other methods. Moreover, the researchers do not know PALS and do not understand the results. If I'm wrong, please give one example of a very significant result that only PALS achieved. (The result of free volume in molecular solids is well known.)

Author Response

Dear Reviewer,

Thank you for the valuable comments, which provided a useful means for improving our review. We accepted the comments and modified the paper accordingly. 

Yours sincerely,

Romána Zelkó

Reviewer 3 Report

The authors present a literature search related to positron annihilation lifetime spectroscopy (PALS). In the present form, the manuscript is not suited for publication for several reasons:

--The paper does not contain new insights.

--The scientific language is not appropriate and the text reads cumbersome.

--The text is often misleading.

--The paper contains several basic errors.

Obviously, the authors are not experts in the field of positron annihilation spectroscopy at all. 

Exemplarily, I just comment in detail the first paragraph - i.e. the first two phrases, l. 34-38 - of the introduction, which reads cumbersome anyway:  “…PALS is a non-destructive and effective structural testing  method  of  materials…” This is misleading if not false: In contrast to this statement PALS is not applied to reveal the structure as it is done by, e.g., diffraction methods. PALS is basically used to analyze vacancy-like defects in crystals or the free volume in amorphous matter. ”…  which  is  based  on  the  fact  that  the  irradiated positron  antiparticle  interacts  also  with  the  electrons  of  the  free  volume  holes  in  the sample.” What is irradiated positron? This should be rewritten.  “The decay of the ortho-Positronium (o-Ps) is accompanied by photon emission…” Radioactive atoms are decaying; Ps annihilates.

In the next two paragraphs (l. 39-57) the authors list many examples; references, however, are missing.

In the following, again, it becomes obvious that the authors just paraphrase any text from text books or other articles and turn the phrases worse or even physically wrong. Even on Table 1 contains basic physical errors; e.g., the vacuum lifetime of p-Ps is 0.125 ns. In the introduction, the authors only refer two references; it seems that they used this as a basis to rewrite their content.

If the authors should decide to submit a completely revised manuscript I strongly recommend to take advice from a positron expert.

Author Response

Dear Reviewer,

Thank you for the valuable comments, which provided a profound initiative for improving our paper. We accepted all the comments and modified the paper accordingly.

Yours sincerely,

Round 2

Reviewer 1 Report

No.

Reviewer 2 Report

I can express my opinion that such a review does not improve our knowledge about the positron annihilation in matter.

Reviewer 3 Report

The revised version can now be published. Final careful text editing is recommended.